



# Retrieval of SWE from dual-frequency radar measurements: Using timeseries to overcome the need for accurate a priori information

Michael Durand[1], Joel T. Johnson[2], Jack Dechow[1], Leung Tsang[3], Firoz Borah[3], and Edward J. Kim[4]

[1]School of Earth Sciences, Ohio State University, Columbus, Ohio 43210, USA
[2]Department of Electrical and Computer Engineering, Ohio State University, Columbus, Ohio 43210, USA
[3]University of Michigan, Ann Arbor, Michigan 48109, USA
[4]NASA Goddard Space Flight Center, Greenbelt, Maryland, 20771, USA

**Correspondence:** Michael Durand (durand.8@osu.edu)

**Abstract.** Measurements of radar backscatter are sensitive to snow water equivalent (SWE) across a wide range of frequencies, motivating proposals for satellite missions to measure global distributions of SWE. However, radar backscatter measurements are also sensitive to snow stratigraphy, microstructure and to surface roughness, complicating SWE retrieval. A number of recent advances have created new tools and datasets with which to address the retrieval problem, including a parameterized

relationship between SWE, microstructure, and radar backscatter, and methods to characterize surface scattering. Although many algorithms also introduce external (prior) information on SWE or snow microstructure, the precision of the prior datasets used must be high in some cases in order to achieve accurate SWE retrieval.

We hypothesize that a time series of radar measurements can be used to solve this problem, and demonstrate that SWE retrieval with acceptable error characteristics is achievable by using previous retrievals as priors for subsequent retrievals. We

demonstrate the accuracy of three configurations of the prior information: using a global SWE model, using the previously retrieved SWE, and using a weighted average of the model and the previous retrieval. We assess the robustness of the approach by quantifying the sensitivity of the SWE retrieval accuracy to SWE biases artificially introduced in the prior. We find that the retrieval with the weighted averaged prior demonstrates SWE accuracy better than than 20%, and an error increase of only 3% relative RMSE per 10% change in prior bias; the algorithm is thus both accurate and robust. This finding strengthens the case

for future radar-based satellite missions to map SWE globally.

## 1 Introduction

Snow water equivalent (SWE) is an important component of the global cryosphere, but is poorly measured globally (Mortimer et al., 2020). Multiple spaceborne satellites have been proposed by space agencies to observe global SWE, but to date none have been selected or launched (Tsang et al., 2022). One often-cited reason for non-selection has been a stated need for high

accuracy a priori information that in practice is challenging to obtain (Rott et al., 2012).

Radar backscatter measurements are sensitive to SWE, but are also impacted by other environmental parameters, such as forest canopies (Lemmetyinen et al., 2022), snow microstructure (King et al., 2018; Rutter et al., 2019; Sandells et al., 2021) and soil moisture and roughenss (Zhu et al., 2022). These "nuisance parameters" motivate the introduction of a priori information





to help constrain SWE retrieval (Tsang et al., 2022). Indeed, a priori information is pivotal for the retrieval algorithms of many

satellite missions. For example, prior information is critical to estimation of river discharge from the recently-launched Surface Water and Ocean Topography satellite mission (Durand et al., 2023). However, it is vital that studies characterize the sensitivity of retrievals to priors.

Recent advances have helped to clarify the relationship between radar backscatter and snow properties. Microcomputed tomography provides a means to objectively characterize snow microstructure, and thus examine its effects on microwave

scattering (Sandells et al., 2021; Picard et al., 2022). These fundamental advances enable new physically-based retrieval studies. For example, Pan et al. (2023) used a two-layer physically-based radiative transfer model coupled with an iterative Markov Chain Monte Carlo methodology that was able to accurately estimate soil properties, SWE, and snow microstructure using in situ measurements of radar backscatter. The study further demonstrated that SWE could be retrieved even in the presence of biases in the prior information. However, the algorithm was quite computationally expensive, making it less suitable for

satellite applications.

A number of recent advances have created new tools and methods with which to address the SWE retrieval problem. Zhu et al. (2018) presented methods to separate snow volume scattering from surface scattering by differencing the radar backscatter on two different days, thus accounting for the effect of the surface scattering on the radar signal. In order to reduce the total number of unknowns in the retrieval problem, Cui et al. (2016) and Zhu et al. (2021) fit empirical relationships to radiative

transfer model simulations for complex snow media (Xu et al., 2012). Zhu et al. (2021) present such a model in which snow volume backscattering and attenuation of the surface scattering by the snowpack are parameterized by the SWE and single-scattering albedo $\omega$ at one frequency. These two advances together help to reduce the number of unknowns in the retrieval problem, thus making global SWE retrievals more feasible for future satellite missions.

Additional advances have been published that investigate the application of a priori information for SWE retrievals. Some

past studies have indicated that prior datasets must be highly precise in order to achieve accurate SWE retrieval. The CoReH2O satellite mission, for example, required a high precision prior estimate of snow grain size Rott et al. (2012). Similarly, Rutter et al. (2019) found that grain size would need to be known to within 10 % to enable accurate SWE retrievals. Recently, Zhu et al. (2018) analyzed this problem in terms of the single-scattering albedo $\omega$, and found in airborne datasets that the associated $\omega$ values were grouped into discrete sets of values. Thus, Zhu et al. (2018) recast the need for a priori information on $\omega$ into

a classification problem. The best choice among the discrete possible values of $\omega$ is determined using an a priori estimate of SWE and the measured backscatter. This "$\omega$ classification" approach was successfully used by Zhu et al. (2021) with in situ measurements. This approach simplifies the problem into needing only to know the classified $\omega$ value, which can be determined from prior information on SWE.

In this study, we deploy the parameterized model and retrieval algorithm of Zhu et al. (2021), including background subtrac-

tion and $\omega$ classification, and assess its robustness to a priori SWE accuracy. We hypothesize that using the radar observation timeseries significantly lessens the impact of a prior information, and explore using the previous SWE retrieval as the prior for the subsequent retrieval estimate. The goal is to demonstrate accurate SWE retrievals from radar timeseries measurements that are robust to the accuracy of a priori information in order to strengthen the case for future radar-based satellite SWE missions.



## 2   Datasets and Study Area

We use data from the Nordic Snow Radar Experiment (NoSREx) to explore this hypothesis (Lemmetyinen et al., 2016). We use data spanning the winters ending in 2010 and 2011 and refer to each winter by the year in which it ends. We use tower-based and in situ data from the NoSREx Intensive Observation Area (67.362 °N, 26.633 °E), located at the Finnish Meteorological Institute Arctic Research Centre at Sodankylä, Finland. The SnowScat multi-frequency scatterometer measured radar backscatter in a clearing of a typical boreal forest, with in situ snow measurements and meteorology measurements made
in adjacent areas.

### 2.1   SnowScat scatterometer data

The SnowScat scatterometer instrument was installed on a 9.6-m height tower to observe undisturbed, natural snowpack at several incidence and azimuth angle combinations. SnowScat measures $hh$, $vh$, $hv$, and $vv$-polarized radar backscatter in the frequency range from 9.2 to 17.9 GHz every 3 or 4 hours (depending on the year) at four incidence angles. In this study, we
average the data within each day, and use $vv$ polarized data at 40° incidence angle at 10.2 and 16.7 GHz, which we refer to as X- and Ku- bands, respectively. Measurement uncertainty was assessed by repeat measurements of a calibration sphere, and was approximately 1.0 dB (Lemmetyinen et al., 2016).

### 2.2   Snowpit measurements of SWE

Snowpit measurements were made approximately weekly. SWE was assessed at each snowpit via measurement of snow density
at 5 cm vertical intervals using a $250 \, \text{cm}^3$ (Lemmetyinen et al., 2016) snow volume. A total of 17 and 13 snowpit measurements are used for 2010 and 2011, respectively. The snowpit-derived measurements are the most reliable measurement of SWE available at NoSREx.

### 2.3   Gamma SWE

Measurements of SWE were also made using an automated experimental sensor, the Gamma Water Instrument (GWI). The
GWI estimates SWE by measuring the extinction of gamma rays in the snowpack. GWI measurements were made every three or four hours, at the same temporal frequency as the SnowScat radar measurements (Lemmetyinen et al., 2016). These measurements are less precise than the snowpit SWE measurements.

### 2.4   Meteorology

Meteorology measurements at hourly intervals were made at a tower several meters from the radar, snowpit and GWI sensors
(Lemmetyinen et al., 2016). In this study, we use air temperature and precipitation measurements as described in Section 4.2, with air temperatures averaged and precipitation accumulated for each day.





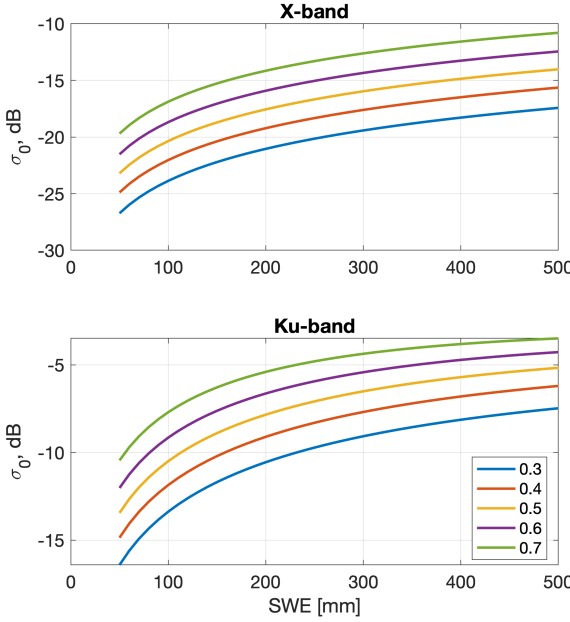

**Figure 1.** The parameterized forward model $M_{vol}(x)$ illustrating the $vv$ polarized normalized radar cross section (NRCS) of snow volume backscattering at 40 degrees incidence in terms of SWE and $\omega$ (values in legend) at 9.2 (upper) and 17.9 GHz (lower).

## 2.5 ERA5 model data

Monthly estimates of SWE are obtained from the ERA5 European Reanalysis, a component of the European ECMWF numerical weather prediction model (Hersbach et al., 2020). As described in Hersbach et al. (2020), ERA5 includes land data
assimilation methodologies that assimilate snow station observations. It is possible that ERA5 is in fact dependent on the NoSREx station data, making it more accurate in our study area than in other areas. In order to address this, we examine the sensitivity of the retrieval algorithm to bias by looking at the sensitivity of the retrieval to a wide range of bias artificially added to the ERA5 data, as described in Section 4.5.

## 3   Retrieval Algorithm Formulation and Application

The retrieval algorithm is a cost function minimization approach, a legacy algorithm with many years heritage (Rott et al., 2012). The minimization approach simply identifies the choice of unknowns (i.e. SWE) to minimize a cost function that includes the difference between observed radar backscatter and model predictions of the same. This section describes the basic formulation of the algorithm, with additional details of how the algorithm is applied for this study (e.g. estimation of surface scattering) described in the subsequent section. The algorithm embeds the parameterized forward model presented by Zhu et al.
(2021), in which snow volume scattering is parameterized as a function of the SWE and the single-scattering albedo at X-band





($\omega$). A set of empirical equations are used to approximate the full response of radar backscatter to snowpack characteristics derived by the more complex bi-continuous Dense Media Radiative Transfer model of Xu et al. (2012) and Ding et al. (2010). Figure 1 shows the model radar predictions as a function of SWE and $\omega$, and shows that rapid increase in backscatter that occurs as snow initially accumulates, as well as the influence of the snow microstructure parametrized by $\omega$.

## 3.1    Formulation

The cost function minimization approach described here builds from the approach of Zhu et al. (2021). The unknowns SWE and $\omega$ are represented as a control vector $x$ and the parameterized model described above is represented as an operator $\sigma_{0,mod}$. In this study, we minimize differences between observations and model predictions in units of decibels [dB], and units of all $\sigma_0$ quantities are in dB unless otherwise noted. The optimal value of the parameters $\hat{x}$ is computed by minimizing :

$$\hat{x} = \arg\min_{x} \left( [\sigma_{0,obs} - \sigma_{0,mod}(x)] \Sigma_{obs}^{-1} [\sigma_{0,obs} - \sigma_{0,mod}(x)]^T + [x - \bar{x}] \Sigma_x^{-1} [x - \bar{x}]^T \right) \tag{1}$$

where $\sigma_{0,obs}$ is the vector of the volume scattering part of the radar observations, $\Sigma_{obs}$ is the error covariance matrix of the observations, $\bar{x}$ is the vector of the prior estimates of SWE and $\omega$, and $\Sigma_x$ is the error covariance of the prior estimates. The parameterized forward model is the sum of the attenuated surface scattering and the parameterized volume scattering, but these quantities are additive only in linear units, not in dB; we use an "$*$" superscript to denote linear quantities. The forward model can be written as:

$$\sigma_{0,mod}^*(x) = f(x)\, \sigma_{0,surf}^* + M_{vol}^*(x) \tag{2}$$

where $\sigma_{0,surf}^*$ is the surface backscatter, $f(x)$ represents the attenuation of the surface backscatter by the snowpack (which depends on SWE and $\omega$ in the parameterized model), and $M_{vol}^*(x)$ is the parameterized snow volume scattering model expressed in linear units.

Following Zhu et al. (2018), we assume that the prior estimate of the single scattering albedo $\bar{\omega}$ is either 0.4 or 0.6 in order to indicate whether snowpack is dominated by large scatterers such as depth hoar. We choose $\bar{\omega}$ using a two step approach. We first solve the optimization problem

$$\hat{\omega} = \arg\min_{\omega} \left( [\sigma_{0,obs} - \sigma_{0,mod}(x_0)] \Sigma_{obs}^{-1} [\sigma_{0,obs} - \sigma_{0,mod}(x_0)]^T \right) \tag{3}$$

in which $x_0$ fixes SWE at its prior estimate while $\omega$ is allowed to vary. We then set $\bar{\omega}$ to be either 0.4 or 0.6, depending on which is closer to $\hat{\omega}$. Following Zhu et al. (2018), the prior covariance matrix $\Sigma_x$ is then assumed diagonal with standard deviation 0.1 for $\bar{\omega}$ and 50% of the ERA5- SWE value for $S\bar{W}E$.

## 3.2    Three configurations for estimating prior SWE and $\omega$

We explore three configurations for determining the prior value for SWE ($\overline{SWE}$) that is then used both in Equation (1) and in selecting $\hat{\omega}$. The first assumes the prior SWE to be equal to the ERA5-Land SWE ($SWE_{ERA}$) at each time step; this is referred to as the "ERA prior" hereafter. The second sets the prior SWE estimate to be equal to the previous SWE retrieval (labeled as





$S\hat{W}E_{t-1}$). For the first retrieval of each year, when there is not yet a "previous" retrieval, the prior SWE is set equal to the ERA5 SWE.

The third configuration uses the weighted average

$$\overline{SWE}_t = g_{ERA}SWE_{ERA} + (1 - g_{ERA})S\hat{W}E_{t-1} \tag{4}$$

where $g_{ERA}$ is the weight given to the estimate of SWE from ERA5. The optimal weight can be expressed in terms of the variances of the respective terms

$$g_{ERA} = \frac{\sigma^2_{S\hat{W}E}}{\sigma^2_{S\hat{W}E} + \sigma^2_{ERA}} \tag{5}$$

where the $\sigma$ terms represent the uncertainty in ERA and the previous SWE estimate, respectively. Assuming that $\sigma_{ERA}$ is 50% of $SWE_{ERA}$ and that the uncertainty of $S\hat{W}E_{t-1}$ is 35% (accounting for the actual accuracy of the previous retrieval and

allowing for the potential of SWE to change between successive retrieval days due to additional precipitation events), then $w_{ERA} = 0.33$, which is the value used in what follows. This approach is essentially an optimal weighting between the ERA5 SWE and the previous retrieval; we refer to it as the "weighted average" hereafter. For the first retrieval, when there is not a "previous" retrieval yet, the prior SWE is set equal to the ERA5 SWE.

For the third configuration, the prior for $\omega$ is similarly computed as a weighted average between the $\omega$ from the previous

retrieval, and the $\omega$ computed based on the ERA5 SWE, using the same weight as used for SWE, and then classified into either a value of 0.4 or 0.6. In other words, we first compute the prior for time $t$ as described in the last paragraph of Section 3.1, which we will refer to as $\omega_{ERA}$. We then calculate a weighted average of that value and the previously retrieved $\omega$:

$$\overline{\omega}_t = g_{ERA}\omega_{ERA} + (1 - g_{ERA})\hat{\omega}_{t-1} \tag{6}$$

where we use the same weight $g_{ERA}$ for $\omega$ as used for SWE. Then this value of $\overline{\omega}_t$ is used to choose the prior for time $t$ of

either 0.4 or 0.6.

## 4 Experiment Design

### 4.1 Estimating surface scattering

We estimate the surface scattering independently for each of the two years in this study using the ERA5 SWE and early-season backscatter measurements. Indeed, the early-season backscatter days have a non-zero SWE that is accounted for by solving

Equation 2 for the surface backscatter:

$$\sigma_{0,surf} = \frac{\sigma_{0,obs,b} - M_{vol}(x_b)}{f(x_b)} \tag{7}$$

where $\sigma_{0,obs,b}$ is the observed backscatter on the day chosen to use for background estimation and $x_b$ uses the ERA5 SWE value and $\omega = 0.5$ (results were similar for $\omega$ between 0.4 and 0.6 due to the low SWE values these times). As shown in Table 1, background NRCS values determined at X-band are fairly consistent between the two years, while Ku-band values show

larger variations.





**Table 1.** $\sigma_{0,surf}$ values determined for the two years analyzed in this study. True SWE is estimated as described in Section 4.2.

| Year | Date | True SWE [mm] | X-band [dB] | Ku-band [dB] |
|------|------|---------------|-------------|--------------|
| 2010 | November 1 | 23 | -20.2 | -17.0 |
| 2011 | November 20 | 46 | -18.7 | -13.3 |

## 4.2 Estimating true SWE

As described in Section 2, two in-situ SWE datasets are available. Snowpit data are fairly infrequent, covering 36 of the total possible 316 nominal days of the study period (all days from December 1 through March 15). The automated GWI data cover the entire period, but have much higher SWE uncertainty, as they were measured by an experimental sensor (Lemmetyinen et al., 2016). To obtain daily SWE estimates, we use the simple data assimilation approach described in Appendix A to merge the snowpit and GWI data with a simple snow model driven by precipitation and temperature. These daily SWE estimates agree with snowpit and GWI data when available, and also agree with the temperature and precipitation data. The output of the data assimilation is referred to as the "true SWE", hereafter, and is used as the basis for evaluating SWE retrievals.

## 4.3 Flagging wet snow

A small amount of liquid water in the snowpack causes radar backscatter to drop. Throughout the dataset, there are occasional excursions related to the presence of liquid water. We flag these events using the simple but effective wet snow flagging algorithm described in Appendix B. The algorithm has a single parameter: the amount that the data drops in the Ku channel from one day to the next when snow is wet. We do not show the backscatter or snow retrieval data when when wet snow is flagged in the main body of the manuscript (all data are shown in Appendix B), and we do not include the flagged data in assessing algorithm accuracy. See Appendix B for further details on the wet snow flagging algorithm.

## 4.4 Estimating overall forward model uncertainty

The uncertainty of the observations, $\Sigma_{obs}$, must also be specified in Equation 1. There are at least three components that contribute to this uncertainty. The first source is the observations themselves. According to Lemmetyinen et al. (2016), measurement uncertainty was approximately 1.0 dB, based on repeat measurements of a radar calibration sphere. Examination of the backscatter data, however, clearly show that the radar measurements are far more precise than 1 dB: e.g. backscatter generally remains stable during the study period between snowfall and melt events. The averaging of the multiple 3 or 4 hourly radar measurements on each date further reduces this uncertainty. The second source is the parameterized model. Snowpack stratigraphy causes vertical variability in snow scatterers, so that representing layered snowpack backscattering using a single layer parameterized model represents a potential error source that has not been well-characterized. The third source of un-





certainty lies in the compensation of surface scattering in which $\sigma_{0,surf}$ is assumed to remain constant throughout the entire winter. The observation uncertainty $\Sigma_{o}bs$ merges all of these sources of uncertainty together.

An analysis was performed to estimate the combined error level. The analysis used the true SWE (described in Section 4.2) and the measured backscatter. We compute estimates of "true" surface scattering following the approach of Section 4.1 but using the "true" SWE to estimate the background scattering, and also compute "optimal" estimates of $\omega$. We then evaluate the

forward model with the true SWE, true surface scattering and optimal $\omega$ and compare the model predictions with the measured backscatter. This gives an overall estimate of the total uncertainty of the modeling chain. The results showed a combined observational uncertainty of approximately 0.75 dB (16.6 %) across both channels and both years. These values are used the analysis. We further present sensitivity studies to the observation error in Appendix C.

### 4.5 Experiments and Assessment

Retrievals are performed for each year from December 1 through March 15 in order to exclude periods of predominantly wet snow in both early and late winter. We also artificially introduce additional biases in the ERA5 prior SWE and examine the impact of the retrieved SWE values. Specifically, we multiply the ERA5 prior SWE by a factor of $1 + f_{bias}$, where $f_{bias}$ ranges from -0.5 to 0.5. The retrieval algorithm has no knowledge of the bias and retains the 50% assumption for the uncertainty in the prior ERA5 SWE as described in Section 3.1. Retrieval accuracy is assessed by both the SWE root mean squared error

(RMSE) and the relative RMSE (rRMSE), where the latter is computed by calculating the root mean square of the relative error (i.e. the actual error divided by true SWE) on each day.

## 5 Results

### 5.1 Results using ERA5 prior

Figure 2 shows the measured backscatter time series, the SWE retrieval, true SWE, and prior SWE, and the estimated $\omega$ for

2010 and 2011, for the ERA5 prior configuration. In December 2010, the backscatter measurements especially at Ku-band are somewhat erratic, leading to somewhat less accurate retrievals, while in January 2010 the SWE measurements and true SWE are both relatively constant. Snowfall events then increase the SWE in February 2010, while backscatter remains roughly constant, indicating a drop in $\omega$. NRCS values then increase in late February and March, resulting in retrieved SWE values that converge to the true SWE by the end of the study period. The retrieved $\omega$ remains near 0.6 throughout the winter. Combining

all retrievals across both 2010 and 2011, the ERA5 prior case has rRMSE of 13.7 % and RMSE of 13.8 mm (Table 2).

While this is good performance, the results using this prior configuration are quite sensitive to any bias in the prior. Figure 3 averages performance for 2010 and 2011, and for both negative and positive bias, producing an average response of the algorithm rRMSE to bias. From Figure 3, the rRMSE for ERA5 increases from 13.7 % to 46 % as the artificially imposed bias increases from 0 to 0.5. We compute the sensitivity of the rRMSE to the prior as (46 %-13.7 %) / 0.5 = 0.65. Thus for every 10





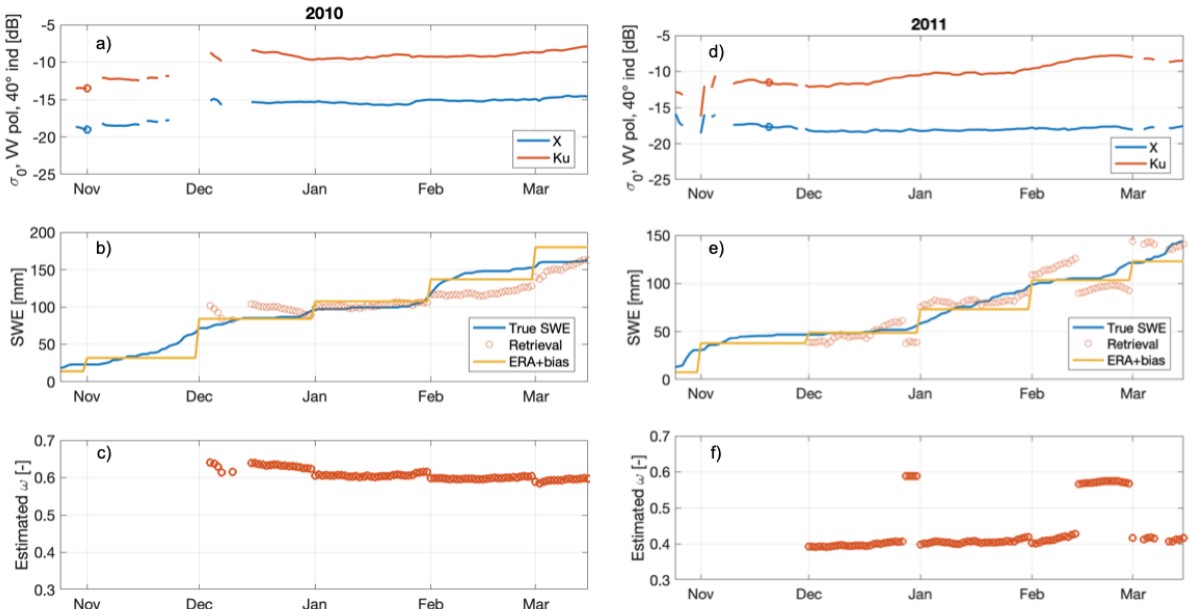

**Figure 2.** For 2010 and 2011, respectively, the observed backscatter (a and d), true, retrieved and prior SWE (b and e) and retrieved $\omega$ (c and f) are shown. The circles in a) and d) show the date selected for characterizing surface scattering for 2010 and 2011, respectively. The results shown here for the prior do not include any artificially imposed bias.

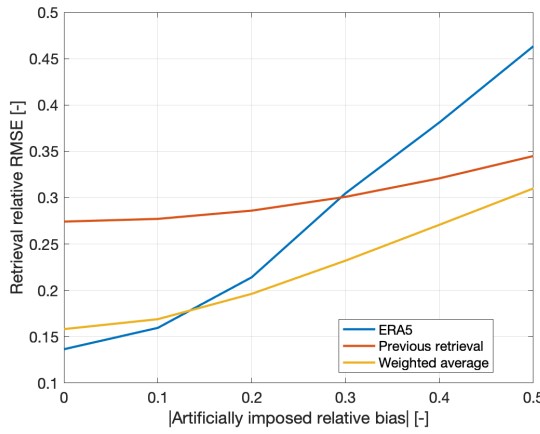

**Figure 3.** Sensitivity of the retrieval relative RMSE to artificially imposed bias on the ERA5 data. These results were obtained by artificially imposing both negative and positive bias, and averaging the resulting relative RMSE

% increase in SWE bias, the rRMSE increases by approximately 6.5 %. Thus the results are quite sensitive to bias in the prior when using the ERA5 data.





**Table 2.** Average between 2010 and 2011 error statistics for the case without any artificially imposed bias for the cost-minimization algorithm for the ERA5 prior and optimal prior, as described above.

| Prior Configuration | rRMSE [%] | RMSE [mm] |
|---|---|---|
| ERA5 | 13.7 | 13.8 |
| Previous Retrieval | 27.4 | 30.1 |
| Weighted Average | 15.8 | 18.9 |

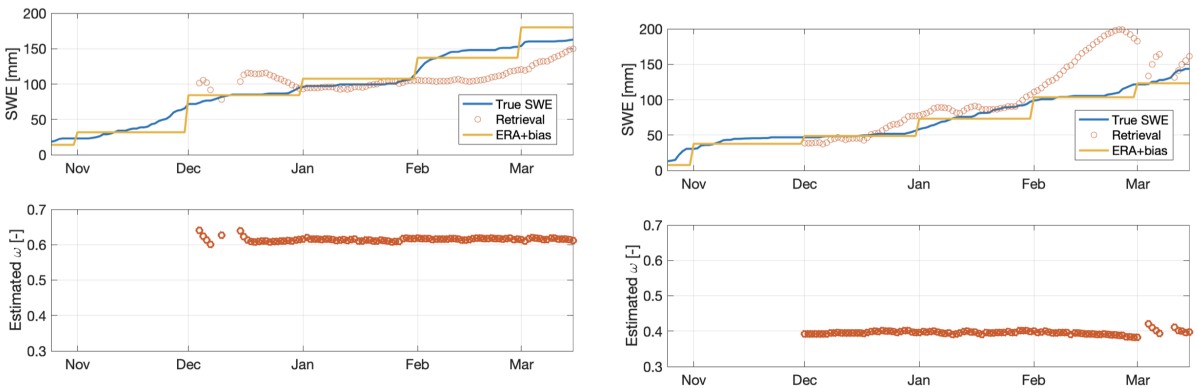

**Figure 4.** As Figure 2, but for the "previous retrieval" prior parameterization.

## 5.2 Results using the "previous retrieval" prior

Figure 4 shows the SWE and $\omega$ retrieval results for the "previous retrieval" prior. In 2010, results are qualitatively fairly similar to those for the ERA5 prior, although the errors in December are larger, and the SWE retrivals remain more constant in early February. In 2011, the $\omega$ retrievals are consistently near 0.4, rather than oscillating between 0.4 and 0.6 in the ERA5 SWE prior. However, the "previous retrieval" results are more distinct from the ERA5 prior case in February 2011, and show a significant overestimation peaking around February 20. As the radar backscatter measurements decreases beginning March 1, the SWE estimates also move back towards the true SWE, and estimates are fairly accurate in March. This divergence in February highlights a weakness of using the previous retrieval. From Table 2, the rRMSE for both years combined is 27.4 %, approximately double that of the ERA5 prior. From Figure 3, however, we see the great advantage of using the "previous retrieval" prior configuration: the SWE estimates are nearly insensitive to bias artificially imposed on the prior SWE estimates from ERA5. The rRMSE increases in Figure 3 from 0.27 to 0.34, a bias sensitivity of 0.14: a 10 % increase in ERA5 prior bias leads to an increase of only 1.4 % in the rRMSE. Thus the ERA5 prior is relatively accurate but not robust to prior bias, while the previous SWE retrieval is relatively inaccurate, but quite robust, further motivating a weighted combination of both.





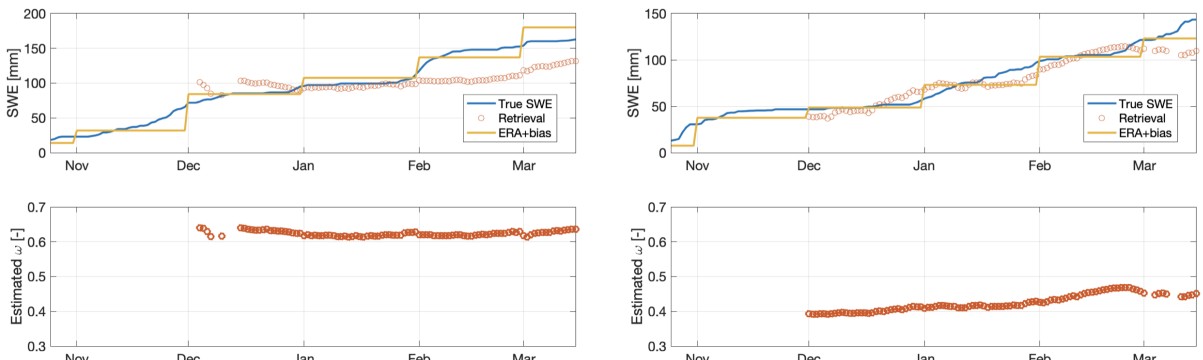

**Figure 5.** As Figure 2, but for the "weighted average" prior parameterization.

## 5.3 Results using the "weighted average" prior

Figure 5 shows the SWE and $\omega$ retrieval results for the "weighted average" prior. The results for 2010 are qualitatively quite similar to those for the "previous retrieval" prior parameterization. However, the errors in December are noticeably smaller. The results for 2011 are a significant improvement over either the ERA5 prior or the previous retrieval prior. The $\omega$ values do not oscillate as they do for the ERA5 prior. Retrievals also show improved performance in February of both years. From Table 2, the rRMSE for both years combined is 15.8 %, far better than the previous retrieval configuration, and only marginally larger than the ERA5 prior. From Figure 3, the sensitivity of the "weighted average" SWE retrievals similarly lies between the two other configurations. The relative RMSE increases from 15.8 % to 31 %, an increase of just 15.2 % for the bias increasing 50 %, a sensitivity of 0.3. Thus, the "weighted average" prior represents a compromise between the two other configurations: it is high accuracy but also relatively insensitive to bias in the ERA5 SWE.

## 6 Discussion

Modern retrieval algorithms often leverage a priori information. When they do so, retrieval algorithms must be shown to be both accurate and relatively insensitive to the a priori information. In this study, we analyze three different configurations for a priori SWE. We show that using the ERA5 prior leads to high retrieval accuracy for this study area, but that retrieval results using the ERA5 prior are too sensitive to prior SWE. Thus, using ERA5 or another model alone as a prior is a risky strategy with the algorithm described above. The second configuration uses the previous retrieval instead of ERA5 as the prior. This leads to a very low sensitivity to ERA5 accuracy, but a much lower accuracy. This suggested a hybrid approach: we use a weighted average of the ERA5 and the previous SWE, which is a strategy that is both accurate and robust, as shown in Figure 6.

This study helps to show a way forward for satellite mission proposal algorithms. This study does not solve all issues for spaceborne application, especially issues related to forest cover; instead, we focus on benchmarking a simple algorithm that




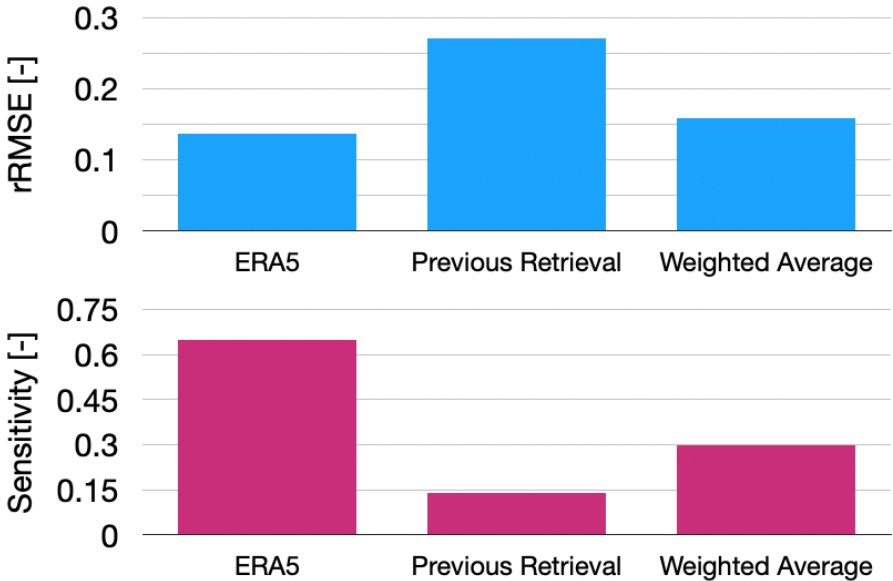

**Figure 6.** Summary of study results: accuracy described by the relative RMSE (top) and sensitivity to the prior estimates of SWE (bottom) for the three different prior configurations.

would function well away from the effects of trees. Regarding measurement frequency, we hypothesize that as long as an observation is available since the most recent significant snowfall event, our "weighted average" algorithm will function with error statistics similar to what are shown here. However, we focus on on the daily backscatter measurements here to isolate the issues of surface scattering and the issue of a priori information on SWE and microstructure. Note that the $\omega$ values in our study

are a proxy for snow microstructure grain size or autocorrelation length (Mätzler, 2002). CoReH2O required a high precision estimate of snow microstructure grain size (Rott et al., 2012). Our "weighted average" algorithm avoids that issue by computing a prior estimate of $\omega$ from the prior SWE. Prior estimates on SWE on widely available, and improving all the time, and our "weighted average" algorithm has a fairly low sensitivity to the accuracy of the model prior of around 0.3. Thus a 10 % bias in the modeled SWE leads to only a 3 % increase in relative RMSE. Other studies such as Zhu et al. (2021) and Lemmetyinen

et al. (2018) use passive microwave measurements to compute either the rough surface scattering or the microstructure grain size correlation length. However, for spaceborne applications, there would be approximately an order of magnitude difference in spatial resolution of passive microwave (on the order of kilometers) and radar (on the order of hundreds of meters), leading to complications. Our approach avoids these issues as well. Finally, the recent study of Pan et al. (2023) shows simultaneous retrieval of SWE and surface scattering using a much more computationally expensive algorithm: approximate differences in

compute time are several orders of magnitude. The Pan et al. (2023) approach is likely more accurate than the one shown here, but at the cost of being potentially too computationally expensive for a global spaceborne application. Our approach solves the most important issues of resolving sensitivity to prior information and surface scattering, while remaining computationally





very low cost. Important next steps for this algorithm include testing on datasets from additional sites, and working on a simple way to track variations in $\omega$, as described in Section D.

## 270  7  Conclusions

This study has applied a previously published algorithm (described by Zhu et al. (2018) and Zhu et al. (2021) and only minimally modified in its application) in order to explore sensitivity to a prior information using in situ radar measurements. A global model (ERA5) was used to obtain prior estimates of SWE; artificial bias was added to the ERA5 SWE estimates in sensitivity experiments. Surface scattering and single-scatter albedo $\omega$ were treated in objective ways based on published

literature requiring no other external or a priori information. We explored three configurations for prior information which use the ERA5 alone, the previous retrieval, and a weighted average of the two, respectively. Using ERA5 alone confirms a problem identified in previous studies: the ERA5 prior SWE must be relatively accurate in order to achieve successful SWE retrievals. However, using the weighted combination of ERA5 and the previous retrieval greatly diminishes the dependence on prior information. This configuration achieved an accuracy of 15.8 % relative RMSE, and increased only 3 % per 10 % increase in

prior bias.

The algorithm presented meets reasonable accuracy requirements, has been shown to be robust to bias in the input prior measurements, and is feasible for implementation in satellite measurements using existing prior datasets. Future work should explore new algorithms employing simple state-space models to track changes in $\omega$ and SWE while preserving the overall simplicity of the approach. These results are limited to shallow snow on flat terrain, with a homogenous footprint, and no

attenuation of the radar measurements by forests. Nonetheless, this study greatly allays a major concern in retrieval of SWE from radar backscatter: the need for accurate prior information. These findings should build confidence in the community that current proposals for future satellite missions will be able to deliver accurate estimates of SWE (Tsang et al., 2022).



## Appendix A:  Estimating True SWE

As described in Sections 2.2 and 2.3, there is no perfect estimate of daily SWE: the snowpit data is relatively infrequent,
but the Gamma SWE measurements are relatively imprecise. We apply a two step process to obtain a daily combined SWE
estimate. In the first step, we use both snowpit and gamma data to calibrate a simple "temperature index" snow model forced
by precipitation and air temperature measurements described in Section 2.4. In the second step, we derive a new estimate
via an optimization algorithm, informed by both the data themselves and the calibrated model. This "true SWE" has superior
agreement with all available snowpit and gamma data, is informed by snowfall timing and air temperature, and is available
daily.

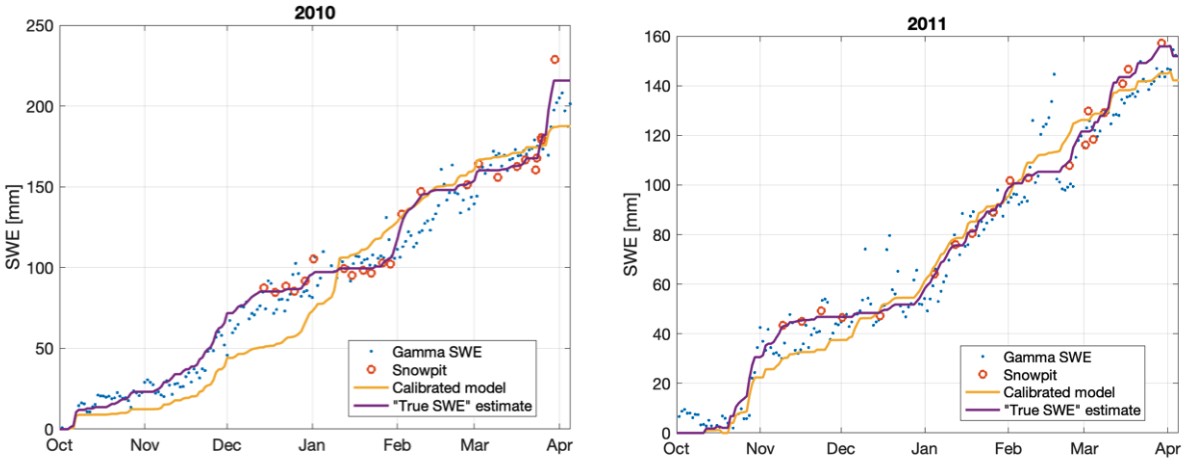

**Figure A1.** Estimates of gamma SWE, snowpit SWE, the calibrated model, and true SWE for 2010 and 2011.

In the first step, we calibrate the temperature-index model of Slater et al. (2013) using both snowpit and gamma SWE ob-
servations. The snow model has a total of five parameters, including a constant multiplicative "gage undercatch" parameter,
air temperature discriminating rainfall from snowfall, and three parameters governing snowmelt. The model is computed one
"water year" at a time, beginning October 1 and running through September 30, at a daily timestep. The calibration objective
function measures the mismatch between model predictions and in situ snow observations, weighted by the respective uncer-
tainty of each observation type. Here we assume snowpit SWE to have 15 mm uncertainty and gamma SWE to have 30 mm
uncertainty. This model calibration also results in a calibrated estimate of snowfall and snowmelt. The result of the calibration
step are SWE simulations that match both pits and gamma SWE to reasonable precision, but neglect any factors not in the
model, such as sublimation or storm-specific errors in precipitation measurement.

In the second step, we compute the "true SWE" formulated as an optimization problem. We compute an estimate of snowfall,
melt, and SWE subject to the constraint of mass balance at each step, with the objective function penalizing differences from the
model calibrated estimates of snowfall and snowmelt, and also from the snowpit SWE. We add further constraints disallowing





**Table A1.** Difference statistics between gamma SWE and "true SWE": mean and root mean square of the difference and relative difference between gamma and true SWE.

| Year | mean SWE [mm] | RMSE SWE [mm] | mean relative SWE [-] | RMSE relative SWE [%] |
|------|---------------|---------------|------------------------|------------------------|
| 2010 | -3.5 | 10.9 | -3.0 | 9.5 |
| 2011 | 0.1 | 10.0 | 1.0 | 13.6 |

snowfall (and thus increase in SWE) when there is no measured precipitation, and disallowing snowmelt (and thus decrease in SWE) when there is no model calibrated snowmelt.

Figure A1 shows the optimal "true SWE" estimates, along with snowpit, gamma, and calibrated model SWE. The calibrated model sometimes tracks the SWE well and sometimes does not, presumably due to time-varying errors in the precipitation undercatch. The gamma SWE sometimes depart by tens of mm from the true SWE estimate, and at other times are significantly different from the snowpit measurements, such as in February of 2010 and February 2011. Table A1 shows that the relative RMS differences between the gamma SWE and the "true SWE" estimate are 9.5 % in 2010 and 13.6 in 2011; these errors

are of the same order of magnitude as the retrieval errors cited throughout the study. The "true SWE" estimate integrates all information into a single estimate, and thus is used as the reference with which to compare SWE retrievals throughout the study.





## Appendix B: Wet snow flagging

The overall wet snow flag strategy in this paper is to attempt to show results only for dry snow. Thus, our intent is not to develop

and verify an objective wet snow algorithm that can be used in other contexts. From Figure B1, some data points contain sharp drops in backscatter. Most of these occur early in the season, but in 2011, some of them occur in March. The presence of wet snow and these sharp drops in backscatter complicates attempts to identify the surface scattering as well as to estimate SWE.

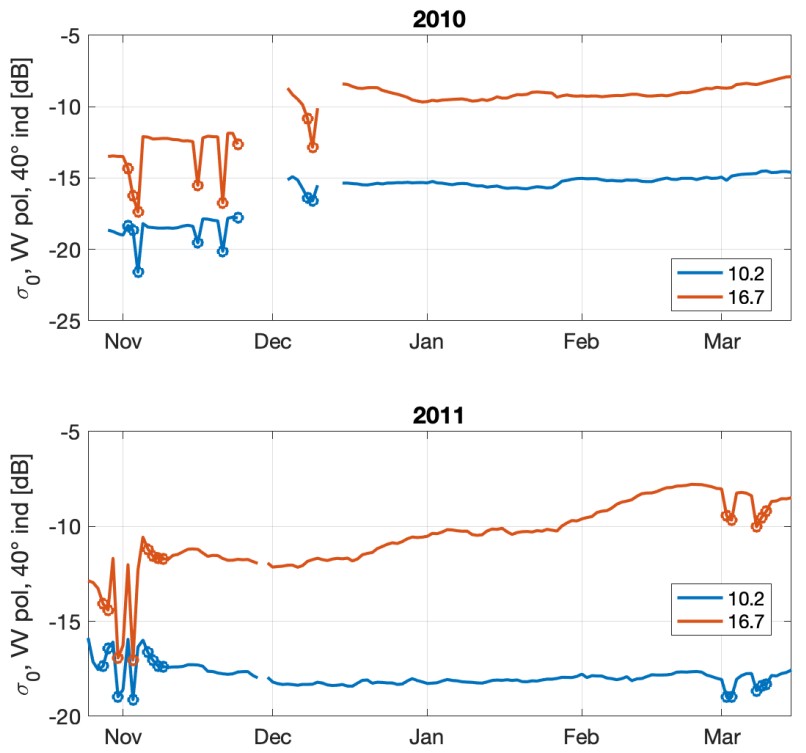

**Figure B1.** The radar data at X- and Ku- band is shown. Data points flagged as "wet snow" are shown as circles.



We analyzed these "sharp drops" for the Ku band data, and found that they represent changes greater than 0.5 dB. We use sharp changes in radar backscatter to compute a wet snow flag. The wet snow flag is set to "true" if the algorithm identifies wet

snow for day $t$.

For the case where the flag is set to "false" on day $t-1$, if the decrease in backscatter from day $t-1$ to day $t$ is greater than 0.5 dB, then we set the flag to "true" for day $t$.

For the case where the flag is set to "true" on day $t-1$, if the increase in backscatter from day $t-1$ to day $t$ is greater than 0.5 dB, then we set the flag to "false" for day $t$.

In other words, the flag stays set as it was on the previous day, unless a sharp change in backscatter causes the flag to change. This simple algorithm identified all of the obviously wet snow in the dataset, failing in only one period of the dataset. In November 2011, the transition from wet snow to dry snow was a gradual increase in backscatter, rather than a sharp rise. We thus added a condition that if the flag stays set at "true" for three consecutive days, we change the flag status to "false".

Further work on more objective and widely applicable algorithms to flag wet snow should be developed. This approach

allows us to remove apparently wet snow data from the analysis, and is thus well-suited to this study.





## Appendix C:  Analysis of observation error

How accurately can the parameterized model explain the measured backscatter data? This question can be addressed (though not definitively answered) by analyzing the model residuals using the true SWE, the "true" estimates of surface scattering, and the "optimal" estimates of $\omega$.

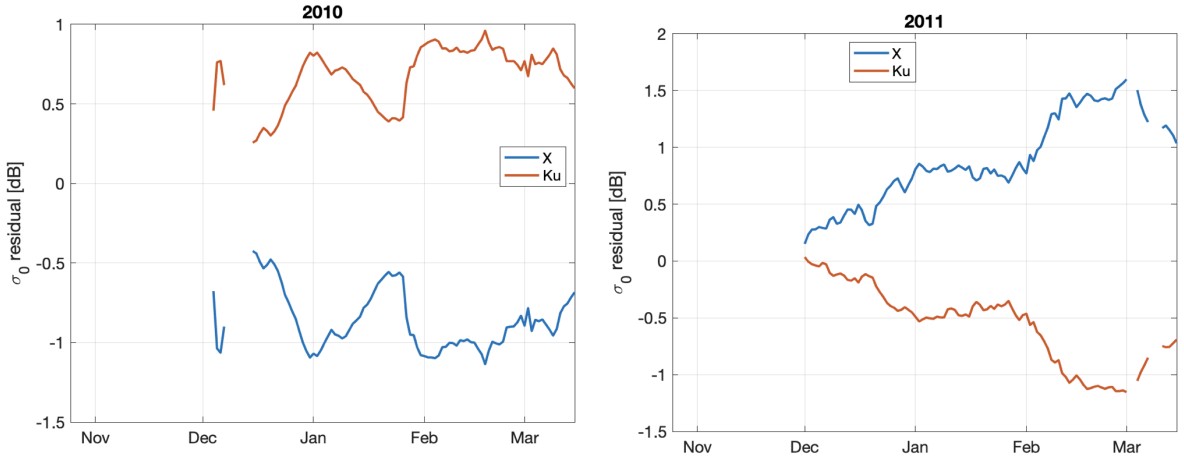

**Figure C1.** Model residuals using true SWE, true surface scattering and optimal $\omega$ for 2010 and 2011.

In both years, the residuals are dominated by bias for both channels: in 2010, Ku residual is always positive and X residual is always negative; the converse is true in 2011. Nearly all of the overall RMS error in both channels and both years is composed of bias (Table C1). The average RMS error for both years and both channels is 0.8 dB.

**Table C1.** Model residual errors: mean (i.e. bias), standard deviation (stdev) and root mean square (RMS) for X and Ku and for both years.

| Year | X mean [dB] | X stdev [dB] | X RMS [dB] | Ku mean [dB] | Ku stdev [dB] | Ku RMS [dB] |
|------|-------------|--------------|------------|--------------|---------------|-------------|
| 2010 | -0.87 | 0.19 | 0.89 | 0.68 | 0.18 | 0.70 |
| 2011 | 0.88 | 0.39 | 0.96 | -0.55 | 0.35 | 0.65 |

From the point of view of SWE retrieval, the practical question is what values of $\Sigma_{obs}$ to use in the cost function (Equation 1). Table C2 shows that the RMS errors that are mostly analyzed in this study are relatively insensitive to the specified observation

errors in $\Sigma_{obs}$.





**Table C2.** Sensitivity of relative RMS SWE errors [%] for the weighted average prior configuration to the specified observation uncertainty standard deviations used in the measurement error covariance matrix $\Sigma_{obs}$.

| Observation Error [dB] | 2010 | 2011 | Average |
|:---:|:---:|:---:|:---:|
| 0.5 | 20.7 | 13.7 | 17.2 |
| 0.75 | 19.1 | 12.5 | 15.8 |
| 1.0 | 18.7 | 13.2 | 15.9 |
| 1.25 | 18.6 | 14.8 | 16.7 |





## Appendix D: Analyzing "optimal" $\omega$ estimates

Despite the "weighted average" algorithm successfully achieving a reasonably high accuracy and low sensitivity to bias, room for improvement remains. For example, Figure 5 shows that the retrieval does not capture the increase in SWE in February 2010. To further investigate the performance of the weighted average approach, we analyzed the "optional" $\omega$ estimates, i.e. values

of $\omega$ that minimize the difference between the parameterized model evaluated at the true SWE. To do this, we additionally calculate a "true" surface scattering estimate, using the true SWE in Equation 7. Using the true surface scattering, true SWE, and observations, we calculate the "optimal" $\omega$ by minimizing the cost function, Equation 3.

    Figure D1 shows the estimates of "optimal" $\omega$ for both years. In 2010, we see drops in $\omega$ near the end of December and the end of January that correspond to increases in SWE, i.e. snowfall events. This dynamic has a simple physical explanation. We

assume that $\omega$ is a metric that can be related to snow microstructure such as a characteristic length scale like the autocorrelation length ($L$) (Mätzler, 2002). As snow ages, $L$ increases due to vapor flux transport as shown in Flanner and Zender (2006). A snowfall event would typically combine new snowfall with smaller $L$ with snowpack with larger $L$, and thus would have the net effect of reducing the combined $L$, as described by (Durand and Margulis, 2006) and (Li et al., 2015). This simple conceptual model explains the lack of change in the measured radar backscatter e.g. in early February 2010: SWE increases,

but a corresponding decrease in $\omega$ cancels any change in measured backscatter. This conceptual model explains 2011 data as well: snowfall events throughout the month of January lead to gradual increase in SWE, corresponding with very little change in measured backscatter and thus a gradual drop in $\omega$. Similarly, snowfall events in late February and early March 2011 are accompanied by a drop in measured backscatter, leading to a drop in estimated $\omega$. Thus the initially counter-intuitive behavior that measured backscatter does not correspond with snowfall events is simply explained by drops in $\omega$ due to new snowfall.

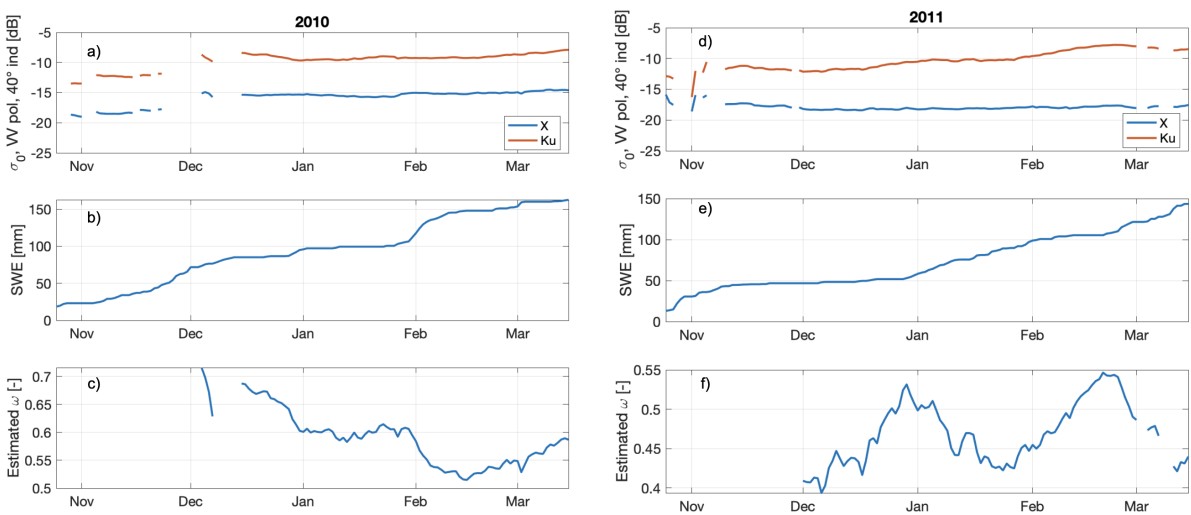

**Figure D1.** Optimal estimates of single scattering albedo ($\omega$), computed as described in the text (c and f) for 2010 and 2011, respectively, along with radar backscatter observations and true SWE.



Figure D1 also shows that increases in measured backscatter occur during periods when SWE is relatively constant. This is consistent with the fact that increases in $L$ drive changes in measured backscatter. In 2010, SWE is fairly constant from mid-February to early March, a period where $\omega$ increases gradually, along with the measured backscatter. In 2011, SWE is constant in December but measured backscatter increases, leading to a corresponding increase in $\omega$. Similarly, $\omega$ increases throughout February, as SWE stays constant and measured backscatter increases throughout the month. Thus the intially counter-intuitive

behavior that measured backscatter increases in periods when SWE is constant is simply explained by increases in $\omega$ due to increasing $L$ due to well-understood microphysical processes.

*Data availability.*   All data are from the NoSREx campaign or ERA5: see Lemmetyinen et al. (2016) or Hersbach et al. (2020), respectively.

*Author contributions.*   Durand wrote the manuscript and performed most of the analyses. Dechow performed some of the analyses. Johnson and Kim advised on the study, and edited the manuscript. Tsang and Borah provided the parameterized model, advised on the study, and

edited the manuscript.

*Competing interests.*   None are present.

*Acknowledgements.*   This work was supported by funding from the NASA Terrestrial Hydrology Program grant 80NSSC17K0200 to Ohio State University.



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
