# Peer review of "Retrieval of SWE from dual-frequency radar measurements: Using timeseries to overcome the need for accurate a priori information"

_EGUsphere, 2023_

## Referee Comment (RC1)

Snow Water Equivalent (SWE) is a key parameter in hydrological, climatological and meteorological applications. New efforts for spaceborne radar-based SWE retrieval algorithms are under development and this paper offers great insight. This paper focuses on the influence of prior information, first guess SWE in this case, on the retrieval. They use previous SWE retrieval in a time series over a winter which reduces the influence of bias from SWE prior coming from an external source. This has benefits in SWE retrieval for future-based satellite missions. This paper is well structure and easy to read. I only have a few comments that would help the understanding of the reader.

Specific comments:

Line 48: the scattering albedo $\omega$ is not well known in the snow community. I suggest defining it a little bit more.

Line 96: "*model predictions of the same*…" what?

Section 4.1: Is it snow surface scattering or ground surface scattering (background)? I think I know the answer, but it is a bit confusing. Sometimes both terms are used (Line 188-189). I suggest sticking to one and defining it more clearly.

Line 173:  remove *when.*

Line 185: "Surface *scattering is assumed to remain constant throughout the entire winter".* Why is that? perhaps cite a paper about constant soil permittivity over the winter.

Line 186: The observation uncertainty symbol is wrong.

Figure 2-3-4: It says on the legend that b) and e) show the SWE of ERA5 + bias. Is it a typo or the bias is indeed shown? It seems like the SWE contains no bias from the curve on the graph.

Section 6: One quick takeaway looking a Fig 2 and 3 is that we don't need retrieval, ERA5 is already good.  ERA5 SWE prior is close to the true SWE, even before the retrieval. I doubt this is a takeaway you want the reader to leave with.  A comment on this was made in section 2.5 on why ERA5 is good at this site but I think commenting again would help. It feels like the method relies on good prior estimation of SWE to predict SWE.

Line 254-255: This information on the scattering albedo could be useful earlier in the intro. Also, this might concern the Zhu et al 2018 retrieval but why not use correlation length or grain size directly as a variable instead of this proxy? I'm not completely sold on the scattering albedo yet!

---

## Author Comment (AC4)

**Further response to reviewers: A possible illustration to better show that we no longer need an accurate prior to retrieve SWE**

Durand, Johnson, Kim, Tsang, Borah, and Dechow

August 25, 2023

**1 Introduction**

In our manuscript, we demonstrate an algorithm that does not need accurate a priori information on either SWE or grain size to accomplish an accurate retrieval. We do this by leveraging the timeseries of radar observations: many retrieval studies try to retrieve each observation independently, which makes the problem more challenging. We use the weighted average of the retrieval from the previous day and an independent model (ERA5 in this study) as the prior.

We have already addressed the reviewers comments in the online discussion in the manuscript: https://egusphere.copernicus.org/preprints/2023/egusphere-2023-1653/ However, we did note in the discussion that we were exploring a possible figure to add that was not explicitly requested by the reviewers, but might help to clear up confusion that they expressed.

In reviewing our manuscript, both reviewers flagged that the fact that the ERA5 modeled SWE is so accurate to begin with may obscure our result. This is a communication issue: in fact our experiments are designed to address this very issue. By varying the bias systematically starting from an accurate prior, we explore in depth the effect of bias on the result, and show that we are not very sensitive to prior bias for the algorithm that takes its prior from a weighted average of the previous retrieval and the model. We propose adding Figure 1 and Figure 2, below. Figure 1 picks two of the many bias scenarios explored in the manuscript and plots them as timeseries. Figure 1 shows visually what 25% bias in the ERA5 SWE looks like. Second, the figure how the SWE retrieval using only ERA5 (triangle markers) is also quite sensitive to prior bias: the biased vs unbiased results are quite different from each other, visually. Finally, the SWE retrieval using the weighted average of the previous retrieval and ERA5 (circle markers) is much less sensitive to the prior: the two retrieval timeseries are not that much different from each other.

To better highlight these differences, we have computed SWE error for the prior and the retrieval results for these two scenarios: unbiased ERA5 and ERA5 with 25% bias artificially added, which is shown in Figure 2. The error

[Figure]

Figure 1: SWE timeseries for 2011 at Sodankyla, showing the "true" SWE computed in the manuscript (blue), the model (ERA5) priors (gold), and the retrievals (red), for two of the bias scenarios studied in the manuscript: no bias, and +25% bias artificially added to the ERA5 model estimates: thicker lines are used to distinguish the results that include bias from those that do not.

values are the difference between the SWE estimate and the "true" SWE at each time. Note that in the retrieval using ERA5 prior (red markers), the retrieval with biased prior (large markers) has much higher error than the retrieval with unbiased prior (small markers). However, in the retrieval using the weighted average (purple markers) the retrieval with biased prior is similar in error to that using the unbiased. Thus, the retrieval using the weighted average prior is much less sensitive to bias in the prior. We attribute this reduced sensitivity to simply using the previous retrieval as the prior: information about the retrieved SWE is thus transmitted from one time to the next, lessening the importance of having an accurate prior input.

In summary, we've presented here a retrieval algorithm that does not require accurate a priori information to accomplish SWE retrievals from radar. We accomplish this by leveraging the timeseries of observations. We believe this has important implications for a long-standing issue in this field: how to retrieve SWE from satellite without having accurate information to start with.

[Figure]

Figure 2: Absolute value of SWE error for the ERA5 model prior (blue
, and three retrieval algorithm configurations (red, gold, and purple) for the
unbiased (thin line and small markers) and scenario with artificial bias added
(thick line and large markers) shown in Figure 1.

---

## Author Response (AR1)

**Response to Reviewers**

Response to reviewers for manuscript egusphere-2023-1653, entitled "Retrieval of SWE from dual-frequency radar measurements: Using timeseries to overcome the need for accurate a priori information", by Durand, Johnson, Dechow, Tsang, Borah and Kim.

Two reviewers have provided excellent comments. Most are very minor, just typos or requests for clarification. Both reviewers made the excellent point that the high accuracy of ERA5 we used as a baseline for the prior creates the inaccurate impression that a high accuracy prior is needed for the retrieval. The paper already contains many places where we explicitly say that the purpose of the paper is exploring that issue, and that our results explicitly show that a high accuracy prior is not needed for accurate SWE retrievals. In order to go even further we added one new figure that includes a new visualization of some of the sensitivity results where we vary the SWE bias systematically and show the response of the three formulations of the retrieval algorithm. While this adds a figure to the manuscript, we believe that it is still a minor change, as it is added only to the discussion and is not a new result, just a new visualization of the same results we had already presented and discussed. The explanation we added of the figure adds additional clarification of the main point of the paper: a high accuracy prior is not needed for accurate SWE retrievals.

Below please find a point-by-point response to the reviewers. We believe the paper is stronger as a result of the reviewers' input. Below we reproduce the editor and review comments in **bold** and our responses in plain type. All of our line numbers below refer to the marked up version of the revised manuscript, in which additions, deletions and replacements are shown in blue font. Note that in the marked up pdf we have provided, we were unable to force our added citations and figure captions to be blue font; we note this where relevant in our responses below.

**Editor**

**No major issues were identified by the reviewers, who kindly offered suggestions to clarify the text. Please proceed to amend your manuscript according to your suggested revisions.**

Thank you. We have responded to all comments below.

**Reviewer 1**

**Snow Water Equivalent (SWE) is a key parameter in hydrological, climatological and meteorological applications. New efforts for spaceborne radar-based SWE retrieval algorithms are under development and this paper offers great insight. This paper focuses on the influence of prior information, first guess SWE in this case, on the retrieval. They use previous SWE retrieval in a time series over a winter which reduces the influence of bias from SWE prior coming from an external source. This has benefits in SWE retrieval for future-based satellite missions. This paper is well structure and easy to read. I only have a few comments that would help the understanding of the reader.**

We thank the reviewer for these comments.

1. **Line 48: the scattering albedo $\omega$ is not well known in the snow community. I suggest defining it a little bit more.**

We agree. We have added lines 42-47, including a definition of the quantity and a citation to where readers can find this definition (Ulaby & Long, 2014).

2. **Line 96: "model predictions of the same…" what?**

Revised (line 103).

3. **Section 4.1: Is it snow surface scattering or ground surface scattering (background)? I think I know the answer, but it is a bit confusing. Sometimes both terms are used (Line 188-189). I suggest sticking to one and defining it more clearly.**

Revised and clarified throughout, including (not limited to) lines 3, 5, 23, 37, 38, 41, 105. Additionally, we added a more explicit definition at line 126-127. We note that "background" is often used in the literature interchangeably with surface backscatter, and so in our definition at lines 126-127, we simply note that the two are used interchangeably.

4. **Line 173: remove when.**

Done, line 181.

5. **Line 185: "Surface scattering is assumed to remain constant throughout the entire winter". Why is that? perhaps cite a paper about constant soil permittivity over the winter.**

We have added a citation of Lemmetyinen et al. (2016b), who shows this; note that the citation was indeed added even though we cannot make the new citation have a blue color in the marked up pdf we provided.

6. **Line 186: The observation uncertainty symbol is wrong.**

Fixed, line 194.

7. **Figure 2-3-4: It says on the legend that b) and e) show the SWE of ERA5 + bias. Is it a typo or the bias is indeed shown? It seems like the SWE contains no bias from the curve on the graph.**

Fixed: replaced figures 2-3-4. These lines do not have bias added.

8. **Section 6: One quick takeaway looking a Fig 2 and 3 is that we don't need retrieval, ERA5 is already good. ERA5 SWE prior is close to the true SWE, even before the retrieval. I doubt this is a takeaway you want the reader to leave with. A comment on**

**this was made in section 2.5 on why ERA5 is good at this site but I think commenting again would help. It feels like the method relies on good prior estimation of SWE to predict SWE.**

Thank you for flagging this. We added a new figure visualizing our bias-perturbed results, and make explicit that our results show that we do not need a prior estimate to accurately retrieve SWE from the radar data: lines 258-271, and added figure 7. Note that figure 7 and its caption are added, even though we were not able to force the caption to be in blue type in the marked up pdf we provided.

9. **Line 254-255: This information on the scattering albedo could be useful earlier in the intro. Also, this might concern the Zhu et al 2018 retrieval but why not use correlation length or grain size directly as a variable instead of this proxy? I'm not completely sold on the scattering albedo yet!**

We have added this information in the introduction: edits run from line 42-46, responding to both this comment and the earlier one on single-scatter albedo.

**Reviewer 2**

**The paper describes a new approach to tackle problems in the retrieval of SWE from radar measurements. Typically high accuracy of a priori SWE and grain size are needed, but the presented algorithm is demonstrated to work with (only) highly-biased SWE as a priori. The results are promising for future snow satellite missions, and the topic is highly relevant. The paper is well-written and easy to read. I have only a few minor comments and suggestions.**

We thank the reviewer for these comments.

1. **P1L13: Remove the duplicate 'than'**

Fixed, line 13.

2. **P1L23: 'Roughenss'**

Fixed, line 23.

3. **P8L192: These values are used \*in\* the analysis.**

Fixed, line 200.

4. **Figs 2, 4, 5: The legend says "ERA+bias", but caption states "do not include any artificially imposed bias". Please check. Should Figs 4 and 5 show the used a priori (previous SWE and weighted average) rather than ERA, or are the legends wrong? Please check.**

Fixed: replaced figures 2-3-4. These lines do not have bias added.

5. **It also looks like the ERA SWE is already very accurate. Why do we need the retrieval? I would highlight the results of Fig. 3 even more to show that retrieval results are good despite biased a priori data. Perhaps add retrievals using biased data to Fig 2,4,5?**

Thank you for flagging this. We added a new figure visualizing our bias-perturbed results, and make explicit that our results show that we do not need a prior estimate to accurately retrieve SWE from the radar data: lines 258-271, and added figure 7. Note that figure 7 and its caption are added, even though we were not able to force the caption to be in blue type in the marked up pdf we provided.

6. **P10L224: "This divergence in February highlights a weakness of using the previous retrieval." Does this mean that if the estimate is wrong, then this error is carried on in further retrievals? Please elaborate.**

Yes, for the second of the three algorithms. The following algorithm fixes exactly this issue. We have added a clarifying comment at line 232-233.

7. **P13 L278: 'dependnence'**

Fixed, line 301.

---

## Author Response (AR2)

**Final Response**

Response to editor for manuscript egusphere-2023-1653, entitled "Retrieval of SWE from dual-frequency radar measurements: Using timeseries to overcome the need for accurate a priori information", by Durand, Johnson, Dechow, Tsang, Borah and Kim.

The editor provided feedback that led to three minor changes.

**Editor**

**Figure 7 – please revisit the choice of colours. In a), do all these need to be orange? It is hard to distinguish between all the symbols. Please also ensure consistency between a) and b)**

**Please correct typo introduced in Conclusions (dependnence)**

This final comment was sent as a private note:

**Just a comment – in appendix D the compensation between single-scatter albedo and SWE is the temporal equivalent of https://agupubs.onlinelibrary.wiley.com/doi/10.1002/2013JF003017 section 4.2. In this paper the lack of spatial variability in measured TB is for the same reason! (no need to change the paper)**

Thank you. We have responded to all comments below.

1. We have replaced Figure 7. Now, all colors are consistent between a) and b). This led to minor changes to the caption as well.
2. We fixed the typo in the conclusions (dependnence)
3. We have added a single sentence, citing this very interesting comparison, which helps make the case that in Appendix D, we are seeing a real outcome, that has been shown previously